# New admissions and asymptomatic TB cases seem to fuel TB epidemic in prisons, a cross sectional survey in Tanzania

Chacha David Mangu[1]*, Petra Clowes[1,2], Jan van den Hombergh[3],
Clement Mwakabenga[1], Simeon Mwanyonga[1], Jane Ambindwile[1], Faith Kayombo[1],
Monica Minja[1], Samuel Kalluvya[4], Lisa Gerwing-Adima[4,5], Christa Kasang[5],
Andreas Mueller[5], Edward Chilolo[3], Juma Angolwisye[6], Dickson Nsajigwa[6],
Adili Kachima[6], Deus Kamala[2], Beatrice Mutayobya[2], Nyanda Elias Ntinginya[1],
Michael Hoelscher[7,8], Elmar Saathoff[7,8], Andrea Rachow[7,8]

1 National Institute for Medical Research, Mbeya Medical Research Center, Mbeya, Tanzania, 2 National TB and Leprosy Program, Dodoma, Tanzania, 3 PharmAccess International, Dar es Salaam, Tanzania, 4 Bugando Medical Centre, Mwanza, Tanzania, 5 Medical Mission Institute, Wuerzburg, Germany, 6 Ministry of Home Affairs, Prison Authority, Dodoma, Tanzania, 7 Division of Infectious Diseases and Tropical Medicine, University Hospital, LMU Munich, Munich, Germany, 8 German Center for Infection Research (DZIF), Partner Site Munich, Munich, Germany

* cmangu@nimr-mmrc.org

**Data Availability Statement:** Prison data in Tanzania are protected by law. The country law requires special permission to be obtain from the

## Abstract

There is an increased risk for tuberculosis (TB) infection and disease progression in prison settings. TB prevalence in prisons in low- and middle-income countries have been measured to be up to 50 -times higher than in the general population. The aim of the study was to perform active TB screening and estimate the burden of TB in central prisons in Tanzania mainland. We performed TB active case finding in five central prisons, Keko, Segerea, Ukonga, Butimba and Ruanda prison in Tanzania, using the Xpert on early morning or spot sputum sample from inmates and new entries between April 2014 and July 2015. A questionnaire that asked about the symptoms and risk factors for TB was administered before a sputum sample was collected. Out of 13,868 incarcerated individuals tested, 13,763 had valid results. TB prevalence among tested was 1550 per 100,000 population (214/13,763); new admissions contributed to the majority (61.68%) of TB cases, but prevalence was higher among inmates (1.75%) compared to new admissions (1.45%). Ukonga, an urban prison which incarcerates long-term convicted inmates had the highest prevalence of 4.02%. Male gender (OR = 2.51, p<0.001), repeated incarcerations (OR = 2.85, p<0.001), history of TB treatment (OR = 1.78, p = 0.002), TB symptoms (OR = 2.78, p = 0.006) and HIV infection (OR = 2.86, p = 0.002) were associated with positive TB results. New admissions could be the driving force of the TB epidemic in the penitentiary system. However, prison environments remain a major risk factor for developing active TB disease.

Ministry of Home Affairs and Prison Authority and the National Ethics Committee before accessing or publishing data related to prison or prisoners in the country. All requests for data should be addressed to: Prison Commissioner General, Prison HQ, Msalato, Arusha Rd. P.O. Box 1176, Dodoma, Tanzania and National Health Research and Ethics Committee, c/o National Institute for Medical Research, 3 Barack Obama Drive P.O. Box 9653, 11101 Dar-es-Salaam Tanzania Email: ethics@nimr.or.tz.

**Funding:** This work took place within the framework of TB screening project in central prisons that was funded by TB REACH grant Wave 3, from the TB Stop Partnership, received by MH. The funders had no role in study design, data collection and analysis, decision to publish, or preparation of the manuscript.

**Competing interests:** The authors have declared that no competing interests exist.

## Introduction

Despite the efforts to reduce the global burden of TB, it has been one of the infectious diseases causing substantial morbidity and mortality especially in Asia and Sub-Saharan Africa [1,2]. It is estimated that 10.6 million people developed TB in 2022, 25% of which occurred in Africa, and caused 1.3 million deaths [3]. TB remains one of the major causes of death among people living with HIV (PLHIV) causing 167,000 deaths in 2022 [3]. TB cases are still under-reported; approximately 7.5 million cases were diagnosed and notified in 2022 which is about 70% of the estimated cases [3]. The majority of missed TB cases most likely occur among people who lack free or direct access to health care services such as persons who experience incarceration. It is estimated that, more than 10.35 million people are held in penal institutions throughout the world in 2023, either as pre-trial detainees/remand or convicted and sentenced [4]. Such individuals therefore are faced with a high-risk environment for new TB infection and fastening the progression to active TB which, among others, is promoted by overcrowding, poor ventilation and poor diet [5,6]. Limited access to health care results in late case detection and delayed treatment initiation with potential poor adherence sustaining the ongoing TB transmission in penal institutions [5,6].

Despite its highly endemic nature, prevalence of TB in prisons is largely unknown in many countries. Systematic reviews showed incidences of 237.6 per 100,000 and 1,942.8 per 100,000 person-years in high and middle/low-income countries, respectively [7,8]. A recently published paper on TB incidences among incarcerated individuals reveal an incidence rate of 1195 per 100,000 person-years in Africa [9]. Other studies have shown TB rates among inmates as high as 5 to 50 times greater than those of the general population across developed and developing countries, respectively [10]. Active mass screening studies in Malawi, Ivory Coast, Zambia, Botswana, Cameroon and Ethiopia, found TB prevalence among prisoners of about 7 to more than 10 times higher than in the general population [11–16]. In Tanzania, there is lack of published information in the prevalence of TB in prisons; a study at Bugando Hospital, Mwanza, showed a proportion of 41% of smear-positive TB cases among prisoners with presumptive TB referred for diagnosis [17].

Despite the high TB prevalence and complexity that exists in the control of tuberculosis in prisons, TB detection in most prisons still rely on symptomatic screening [18]. However, with the availability of approved rapid molecular diagnostic tests such as GeneXpert MTB/RIF assay (Xpert), which are more sensitive and easier to perform, fast systematic TB screening on a large population has been made possible [19]. The aim of this study was to implement a screening strategy to actively detect TB in five central prisons in Tanzania mainland using Xpert and accurately estimate the burden, and understand the related risk factors of TB in prisons.

## Methods

### Prisons and study groups

The project was implemented in five central prisons in Tanzania; Keko, Segerea and Ukonga Prisons within the highly populated business capital of the Indian Ocean coast region Dar es Salaam; Butimba Prison in Northwest region, Mwanza; and Ruanda Prison in the Southern Highland region, Mbeya. All prisons are located in urban settings but admit people from both urban and rural communities. According to the Tanzanian prison authority, during the study period, these prisons housed an average of about 8,000 people in detention and have a turnover of more than 15,000 individuals per year when including the short time remands who are frequently being admitted and discharged. TB screening was done among those who newly

entered the prison, were transferred in from another prison and persons incarcerated present in the prisons at the beginning of the study.

## Study procedures

The project implemented a TB screening protocol between April 2014 and June 2015 combining a screening questionnaire and the rapid molecular diagnostic tool to all the prisoners in five central prisons. All those found in prison on the day screening started were referred to as inmates and those who entered the prison thereafter were referred as new entries. Those who came in from another prison were referred to as transferred in. A screening questionnaire administered by trained prison health staffs was used to collect demographic information, date of entry into prison, history of previous imprisonment, number of inmates sharing a cell, presence of cell mate(s) treated for TB, TB symptoms, self-reported past history of TB treatment, self-reported HIV status and use of anti-retroviral treatment (ART). The WHO recommended standard screening questions about the presence of four symptoms namely cough, fever, night sweats, and loss of weight were used. One spontaneous expectorate (either early morning or spot) sputum sample was collected from all consented incarcerated persons by a trained prison health facility staff, a nurse or laboratory technician. TB diagnosis was done using the GeneXpert MTB/RIF assay (Xpert; Cepheid, Sunnyvale, California, US), which was stationed at each prison health facility and operated by trained laboratory technicians who were also prison staff. In case of a failed test ("error", "invalid" or "no result"), Xpert testing was repeated on the same sample or a newly collected sample if the sample was either inadequate, discarded or of poor quality. For those consented for HIV testing, HIV diagnosis was done according to the national guidelines using SD Bioline HIV-1/2 3.0 (Standard Diagnostics, Inc. Yongin, South Korea), and confirmed by Alere Determine HIV-1/2 (Alere Inc. Massachusetts, USA) and Uni-Gold HIV-1/2 (Trinity Biotech, Co. Wicklow, Ireland).

## Ethical consideration

The study obtained ethical clearance from the National Health Research and Ethics Committee of the National Institute for Medical Research, Tanzania. It was also approved by the Prison Headquarters as well as the National TB and Leprosy Program in Tanzania.

Screening was voluntary; persons incarcerated consented to participate and thumb printed (because by law they were not allowed to sign) the informed consent form before study procedures were done. HIV diagnosis was done through provider-initiated testing and counseling (PITC) by a nurse counselor. Persons incarcerated who did not consent to test for HIV were not excluded from TB screening. Those diagnosed to have TB or HIV were linked to care at the respective prison health facilities. Only those 18 years and above were included in the study.

## Statistical analysis

The main endpoint of this study was bacteriologically confirmed TB diagnosed by Xpert. Univariate analysis was performed to describe the study population of screened prisoners and the prevalence of TB in different strata of the independent variables.

A conceptual frame work (Fig 1) was established to guide multivariable logistic regression analysis in three levels. Variables with a p-value of <0.1 were retained and used to adjust for variables in the subsequent level and retained in the last model; and variables in the last model with the p-value of <0.05 were considered significant risk factors for TB in prisons. Variables considered distal to the outcome and those proximal to the outcome were in the first and last levels respectively. Number of cell mates was forced into the final model due to its importance

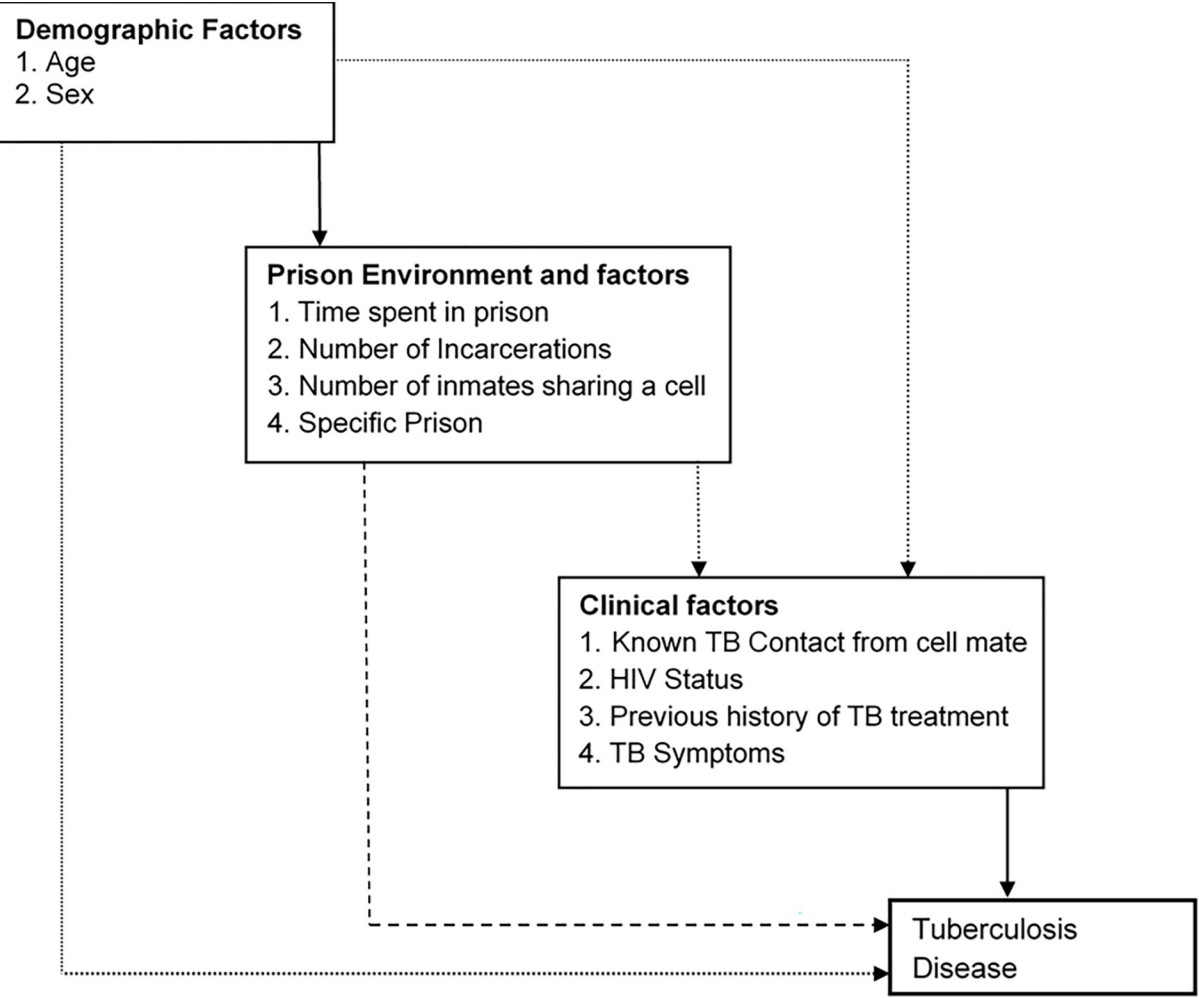

**Fig 1. Hypothesized conceptual framework to guide model building for multivariable analysis.**

in assessing overcrowding within prison cells. Robust standard errors were used in logistic regression to account for correlation within prisons. The Wald test was used to assess significance of the contribution of each variable to the model. Data analysis was performed using Stata/SE version 14 (Stata Corporation, College Station, Texas, US).

## Results

A total of 16,132 prisoners from 5 central prisons agreed to participate in the study and completed the questionnaire. Of these, 13,868 prisoners were screened using Xpert, of which 105 had either an invalid or erroneous final result. Thus, the final analysis included 13,763 prisoners with valid Xpert result (Fig 2).

### Baseline characteristics

Of the 13,763 persons in incarceration, majority (96.3%, n = 13,248) were male. The median age of those screened was 30 years (IQR 24–36 years) while 64.7% were younger than 35 years. A total of 9,077 (66.0%) those screened were new admissions and 12,035 (87.4%) were in prison for the first time. The median time spent in prison was 1.5 months at the time point of

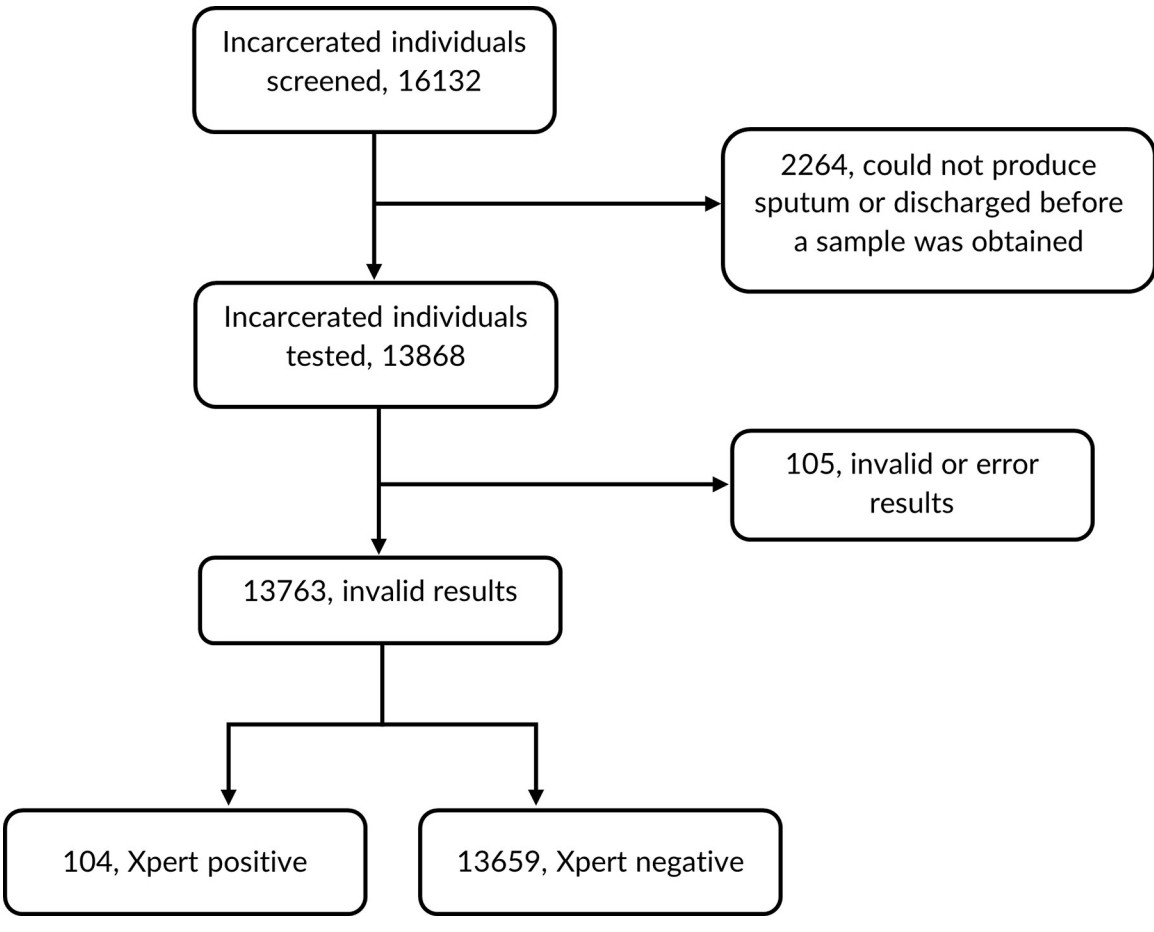

**Fig 2. Flow chart.**

study participation and sharing a cell with more than 10 inmates was common (93.8%, n = 12,911). More than half (51.6%, n = 7,106) had unknown HIV status, and 6.1% (n = 833) were HIV positive (Table 1).

## Risk factors associated with TB diagnosis

In multivariable analysis, males had more than two times the odds for TB infection than females (Odds Ratio [OR] = 2.51, 95% confidence interval [CI] = 1.54 to 4.08, p-value [p] = <0.001). Age did not appear to be an important independent risk factor for active TB. The OR significantly increased with the frequency of incarcerations to nearly three times higher odd for active TB in people who have been incarcerated four or more times (OR = 2.85 CI = 1.78 to 4.55, p<0.001). Previous history of TB treatment was associated with current active TB disease (OR = 1.78, CI = 1.24 to 2.55, p = 0.002). Individuals with TB symptoms were three times more likely to test positive for TB than those without symptoms (OR = 2.78, CI = 1.35 to 5.74, p<0.006) and HIV positive incarcerated individuals also had significantly higher odds of TB than HIV-negative incarcerated persons (OR = 2.86, CI = 1.49 to 5.52, p = 0.002). Being incarcerated in prisons located in the business capital, Ukonga prison or Segerea prison was associated with two times higher OR for having TB (OR = 2.15, CI 1.63 to 2.83, p<0.001 and OR = 2.10, CI 1.75 to 2.52, p<0.001 respectively). Sharing a cell with more than 10 people was not associated with TB infection (Table 2).

**Table 1. Characteristics of the study population and TB cases.**

| Characteristics | Categories | Total (col%) [N = 13,763] | Persons in incarceration with tuberculosis (col%) [N = 214] | Prevalence of Tuberculosis in % | P-value* |
|---|---|---|---|---|---|
| Sex | Female | 515 (3.7) | 4 (1.9) | 0.78 | 0.150 |
| | Male | 13,248 (96.3) | 210 (98.1) | 1.59 | |
| Age in years | <25 | 3,731 (27.1) | 35 (16.4) | 0.94 | 0.004 |
| | 25–34 | 5,177 (37.6) | 87 (40.7) | 1.68 | |
| | 35–44 | 3,030 (22.0) | 58 (27.1) | 1.91 | |
| | >45 | 1,825 (13.3) | 34 (15.9) | 1.86 | |
| Prisoner Status | New Admission | 9,077 (66.0) | 132 (61.7) | 1.45 | 0.184 |
| | Inmate | 4,686 (34.0) | 82 (38.3) | 1.75 | |
| Time Spent in Prison | <6 months | 8,643 (62.8) | 114 (53.3) | 1.32 | 0.023 |
| | 6–24 months | 1,703 (12.4) | 37 (17.3) | 2.17 | |
| | 24–60 months | 1,246 (9.1) | 21 (9.8) | 1.69 | |
| | >60 months | 2,171 (15.8) | 42 (19.6) | 1.93 | |
| Number of Incarcerations | 1 | 12,035 (87.4) | 163 (76.2) | 1.35 | <0.001 |
| | 2–3 | 1,573 (11.4) | 41 (19.2) | 2.61 | |
| | ≥4 | 155 (1.1) | 10 (4.7) | 6.45 | |
| Number of Cell Mates | <5 | 47 (0.3) | 2 (0.9) | 4.26 | 0.195 |
| | 6–10 | 805 (5.9) | 9 (4.2) | 1.12 | |
| | >10 | 12,911 (93.8) | 203 (94.9) | 1.57 | |
| Known TB contact from Cell Mate | No | 13,713 (99.6) | 212 (99.1) | 1.55 | 0.162 |
| | Yes | 50 (0.4) | 2 (0.9) | 4.00 | |
| Previously treated for TB | Yes | 666 (4.8) | 33 (15.4) | 4.95 | <0.001 |
| | No | 13,097 (95.2) | 181 (84.6) | 1.38 | |
| Reported one or more TB symptoms | Yes | 2,261 (16.4) | 92 (43.0) | 4.07 | <0.001 |
| | No | 11,502 (83.6) | 122 (57.0) | 1.06 | |
| HIV Status | Negative | 5,824 (42.3) | 85 (39.7) | 1.46 | <0.001 |
| | Positive | 833 (6.1) | 40 (18.7) | 4.80 | |
| | Not Known | 7,106 (51.6) | 89 (41.6) | 1.25 | |
| Prison ID | Ruanda | 4520 (32.8) | 42 (19.6) | 0.93 | <0.001 |
| | Butimba | 4328 (31.5) | 48 (22.4) | 1.11 | |
| | Ukonga | 828 (6.0) | 32 (15.0) | 3.86 | |
| | Keko | 1921 (14.0) | 26 (12.2) | 1.35 | |
| | Segerea | 2166 (15.7) | 66 (30.8) | 3.05 | |

*chi squared test p-value for difference in TB-prevalence between strata.

## Discussion

Our findings show a high TB burden in Tanzania prisons. The prevalence of bacteriologically confirmed TB of 1,550 cases per 100,000 population reported in our study is 5 times higher than the prevalence in the general population of 295 per 100,000 [20].

Our study contributes important findings over the containment of asymptomatic TB cases in prison settings. Studies have shown that asymptomatic TB cases have the ability to infect others; therefore, sustained and prolonged exposure to TB bacilli could be maintained in prison walls by asymptomatic cases due to missed cases and therefore potentially propagate TB transmission among inmates if conventional symptomatic TB screening is the only screening method relied [17,21–23]. Similarly, in our study we could have missed 57% of TB cases if

**Table 2. Association of various factors with TB disease (N = 13,763; n = 214 (1.555%): Uni- and multi-variable logistic regression results; using robust variance estimates adjusted for clustering by PrisonID.**

| Covariate | N | n | % | Univariable | | | Multivariable | | |
|---|---|---|---|---|---|---|---|---|---|
| | | | | OR | 95% CI | p-value | OR | 95% CI | p-value |
| Sex | | | | | | | | | |
| Female* | 515 | 4 | 0.777 | 1.00 | - | - | 1.00 | - | - |
| Male | 13,248 | 210 | 1.585 | 2.06 | (1.03 to 4.11) | 0.041 | 2.51 | (1.54 to 4.08) | <0.001 |
| Agegroup | | | | | | | | | |
| < 25 yrs* | 3,731 | 35 | 0.938 | 1.00 | - | - | 1.00 | - | - |
| 25 - <35 yrs | 5,177 | 87 | 1.681 | 1.80 | (1.49 to 2.19) | <0.001 | 1.31 | (0.97 to 1.78) | 0.083 |
| 35 - <45 yrs | 3,030 | 58 | 1.914 | 2.06 | (1.33 to 3.20) | 0.001 | 1.28 | (0.75 to 2.21) | 0.365 |
| > = 45 yrs | 1,825 | 34 | 1.863 | 2.00 | (0.98 to 4.11) | 0.058 | 1.30 | (0.70 to 2.42) | 0.399 |
| Number of Incarcerations | | | | | | | | | |
| 1* | 12,035 | 163 | 1.354 | 1.00 | - | - | 1.00 | - | - |
| 2–3 | 1,573 | 41 | 2.606 | 1.95 | (1.15 to 3.31) | 0.013 | 1.57 | (1.17 to 2.09) | 0.002 |
| ≥4 | 155 | 10 | 6.452 | 5.02 | (2.57 to 9.81) | <0.001 | 2.85 | (1.78 to 4.55) | <0.001 |
| Time spent in Prison | | | | | | | | | |
| < = 6 months* | 8,643 | 114 | 1.319 | 1.00 | - | - | 1.00 | - | - |
| 6–24 months | 1,703 | 37 | 2.173 | 1.66 | (1.35 to 2.05) | <0.001 | 1.31 | (1.07 to 1.62) | 0.010 |
| 24–60 months | 1,246 | 21 | 1.685 | 1.28 | (0.79 to 2.09) | 0.318 | 0.87 | (0.61 to 1.22) | 0.414 |
| >60 months | 2,171 | 42 | 1.935 | 1.48 | (0.72 to 3.05) | 0.292 | 0.95 | (0.77 to 1.18) | 0.669 |
| HIV Result | | | | | | | | | |
| Negative* | 5,824 | 85 | 1.459 | 1.00 | - | - | 1.00 | - | - |
| Positive | 833 | 40 | 4.802 | 3.41 | (1.68 to 6.89) | <0.001 | 2.86 | (1.49 to 5.52) | 0.002 |
| Unknown | 7,106 | 89 | 1.252 | 0.86 | (0.67 to 1.09) | 0.214 | 1.18 | (0.86 to 1.61) | 0.314 |
| History of TB Treatment | | | | | | | | | |
| No* | 13,097 | 181 | 1.382 | 1.00 | - | - | 1.00 | - | - |
| Yes | 666 | 33 | 4.955 | 3.72 | (2.87 to 4.82) | <0.001 | 1.78 | (1.24 to 2.55) | 0.002 |
| Number of Cell Mates | | | | | | | | | |
| <10* | 852 | 11 | 1.291 | 1.00 | - | - | 1.00 | - | - |
| >10 | 12,911 | 203 | 1.572 | 1.22 | (0.89 to 1.68) | 0.216 | 0.93 | (0.74 to 1.18) | 0.570 |
| Any TB Symptoms | | | | | | | | | |
| No* | 11,502 | 122 | 1.061 | 1.00 | - | - | 1.00 | - - | - |
| Yes | 2,261 | 92 | 4.069 | 3.96 | (2.13 to 7.35) | <0.001 | 2.78 | (1.35 to 5.74) | 0.006 |
| PrisonID | | | | | | | | | |
| Ruanda* | 4,520 | 42 | 0.929 | 1.00 | - - | - | 1.00 | - - | - |
| Butimba | 4,328 | 48 | 1.109 | 1.20 | (1.20 to 1.20) | <0.001 | 1.10 | (1.02 to 1.20) | 0.019 |
| Ukonga | 828 | 32 | 3.865 | 4.29 | (4.29 to 4.29) | <0.001 | 2.15 | (1.63 to 2.83) | <0.001 |
| Keko | 1,921 | 26 | 1.353 | 1.46 | (1.46 to 1.46) | <0.001 | 1.08 | (1.01 to 1.15) | 0.027 |
| Segera | 2,166 | 66 | 3.047 | 3.35 | (3.35 to 3.35) | <0.001 | 2.10 | (1.75 to 2.52) | <0.001 |

N = number of observations; n = number of positives; % = percent positive; 95% CI = 95% confidence interval.

* reference stratum.

symptomatic screening was the only screening method relied on. For this reason, robust and effective approaches using effective diagnostic tools for TB screening in prisoners are required. Nonetheless, TB symptoms remain a major predictor of TB disease as shown by a TB high prevalence among persons incarcerated with symptoms compared to those without (4.07% vs 1.06% respectively). Therefore, an effective screening intervention is the one that apply mixed methods that detect both symptomatic and asymptomatic TB.

Despite high TB prevalence among persons in detention, the fact that 61% of detected TB cases in our study were among new admissions suggests that new admissions bring TB into the prison facilities and therefore contribute significantly to the TB epidemic in prisons. These findings also observed in other studies [24,25] in countries with high TB burden suggests two possible explanations; firstly, many of those who commit offenses and end up in a penal institution come from a background of high risk for TB infection and therefore bring with them an increased risk for TB infection into prison and actively contribute to the high prevalence of TB in prisons [6]. And secondly, it is highly likely that, specific conditions in prisons not only lead to a higher risk of TB infection but also accelerate progression from latent TB to active disease within a short time. Many TB control strategies in prisons focus on already incarcerated individuals while little or no attention is given to new admissions. Findings from our study suggest that equal weight has to be given to both groups of people who experience detention. Constant screening at prison-entry aims at detecting untreated admissions and thereby reduce introduction of new TB cases into prison while mass and contact screening should aim at detecting the circulating TB within the prison walls [26].

Our findings do not suggest overcrowding to be a risk factor for TB disease in prison; however, this might be due to the possibility that the disaggregation by number of inmates per cell during data collection was not specific enough and hence it probably obscures the association. Similarly, having a known TB contact within the cells was not associated with TB infection in our study, however, this finding could be due to the fact that more than 50% of the identified TB cases were new entries who didn't have prolonged contact with other inmates during the time of screening. Yet, screening of individuals who have contact with index cases should still continue as recommended because it is effective in increasing case detection [26–29]. HIV remains an important single predictor of TB infection in prison settings as it is well known to fuel the TB epidemic [30]. Notably, effective control of TB infection also requires appropriate interventions against HIV infection.

As limitations, using the prison staff in the study could have breached autonomy and confidentiality of study participants. However, to minimize this effect, GCP training was done and frequent supportive supervisions were performed. The study itself was an implementation of screening strategy that aimed at improving capacity for TB screening in prison including building capacity for healthcare workers and hence they had to be used.

## Conclusion

Our study showed a high TB burden in Tanzanian prisons. The majority of positive TB tests came from among symptomatic prisoners, new entries and person who experience incarceration only for a short time in prison. Other TB risk factors were not prison-associated. Persons experiencing incarceration remain a high-risk population; great mobility due to perpetuating referrals between prisons and interactions with the outside communities can increase transmission; our results suggest that TB screening measures within prisons need to be sensitive, robust to capture asymptomatic cases, and with a fast enough turn-over time to combat TB among prisoners and consequently interrupt TB transmission not only in penal institutions but also in the surrounding communities.

TB screening on admission or exit for both for those in detention or incarceration using WHO recommended PCR based rapid diagnostics such as GeneXpert was not a routine practice in Tanzania prisons at the time of this study. However, because of the findings of this study, active TB screening on all prison entry points have been implemented in Tanzania ever since. Future research is needed to assess the impact and cost-effectiveness of the implementation of WHO approved rapid TB diagnostic tools in prison settings.

## Acknowledgments

We acknowledge the effort from all collaborating institutions implementing the study, NIMR-Mbeya Centre, Tanzania (project lead); PharmAccess International, Tanzania; Medical Mission Institute of Wuerzburg, Germany; Bugando Referral Hospital, Tanzania; Division of Infectious Diseases and Tropical Medicine, University of Munich (LMU) Germany (grant holder). We thank the Tanzania Prison Authority Headquarter, and the National TB and Leprosy Program for facilitating study activities. To all the prison health staffs for your dedication and prisoners for your participation, we thank you.

## Author Contributions

**Conceptualization:** Chacha David Mangu, Petra Clowes, Jan van den Hombergh, Michael Hoelscher, Elmar Saathoff, Andrea Rachow.

**Data curation:** Chacha David Mangu, Petra Clowes, Elmar Saathoff.

**Formal analysis:** Chacha David Mangu, Elmar Saathoff, Andrea Rachow.

**Funding acquisition:** Chacha David Mangu, Petra Clowes, Nyanda Elias Ntinginya, Michael Hoelscher, Andrea Rachow.

**Investigation:** Chacha David Mangu, Petra Clowes, Jan van den Hombergh, Clement Mwakabenga, Simeon Mwanyonga, Jane Ambindwile, Faith Kayombo, Monica Minja, Samuel Kalluvya, Lisa Gerwing-Adima, Christa Kasang, Andreas Mueller, Edward Chilolo, Juma Angolwisye, Nyanda Elias Ntinginya, Michael Hoelscher, Andrea Rachow.

**Methodology:** Chacha David Mangu, Petra Clowes, Jan van den Hombergh, Clement Mwakabenga, Simeon Mwanyonga, Jane Ambindwile, Faith Kayombo, Monica Minja, Samuel Kalluvya, Lisa Gerwing-Adima, Christa Kasang, Andreas Mueller, Edward Chilolo, Juma Angolwisye, Deus Kamala, Beatrice Mutayobya, Nyanda Elias Ntinginya, Michael Hoelscher, Andrea Rachow.

**Project administration:** Chacha David Mangu, Petra Clowes, Jan van den Hombergh, Lisa Gerwing-Adima, Christa Kasang, Edward Chilolo, Dickson Nsajigwa, Adili Kachima, Deus Kamala, Beatrice Mutayobya, Nyanda Elias Ntinginya, Michael Hoelscher, Andrea Rachow.

**Resources:** Chacha David Mangu, Petra Clowes, Jan van den Hombergh, Michael Hoelscher, Andrea Rachow.

**Supervision:** Chacha David Mangu, Petra Clowes, Jan van den Hombergh, Simeon Mwanyonga, Samuel Kalluvya, Lisa Gerwing-Adima, Christa Kasang, Michael Hoelscher, Andrea Rachow.

**Validation:** Chacha David Mangu, Petra Clowes, Jan van den Hombergh.

**Writing – original draft:** Chacha David Mangu.

**Writing – review & editing:** Chacha David Mangu, Petra Clowes, Jan van den Hombergh, Clement Mwakabenga, Simeon Mwanyonga, Jane Ambindwile, Faith Kayombo, Monica Minja, Samuel Kalluvya, Christa Kasang, Andreas Mueller, Edward Chilolo, Juma Angolwisye, Dickson Nsajigwa, Adili Kachima, Deus Kamala, Beatrice Mutayobya, Nyanda Elias Ntinginya, Michael Hoelscher, Elmar Saathoff, Andrea Rachow.

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
