## [Decision Letter · Decision Letter 0]

10 Oct 2023

PGPH-D-23-01618

New Admissions and Asymptomatic TB Cases fuel TB Epidemic in Prisons, a Cross Sectional Survey in Tanzania

Dear Dr. Mangu,

Thank you for submitting your manuscript to PLOS Global Public Health. After careful consideration, we feel that it has merit but does not fully meet PLOS Global Public Health’s publication criteria as it currently stands. Therefore, we invite you to submit a revised version of the manuscript that addresses the points raised during the review process.

EDITOR:

1. Was there a specific reason for obtain inform consent by thumb print and not signature? What was the literacy level/ education (needs to be included in table 1) of the study participants? If not all study participants were literate how was informed consent administered. What happened to participants who were diagnosed to have TB?

2. Please clarify why only 1 sputum sample (not 2 samples) was used for diagnosis. Was an attempt made for radiological diagnosis in those symptomatic but unable to produce sputum? Was any procedure used to induce sputum. The study seems to focus on pulmonary TB only, was an attempt to identify extra pulmonary TB made? The prevalence should specify 'pulmonary TB'. How was asymptomatic TB diagnosed? Kindly provide the definition of asymptomatic TB, was this also restricted to only pulmonary TB. Was an attempt to diagnose drug resistant TB made?

3. What is the standard procedure for diagnosis and management of TB in prisons in Tanzania.

4. Was screening and diagnosis done on the day of admission to the prison or after a few days. If after a few days, were any strategies used for screening for TB in contacts within the prison-and outside as applicable. 

5. Kindly add a section on the limitations of your study (primarily methodological issues) at the end of the discussion.

We look forward to receiving your revised manuscript.

Kind regards,

Rashmi Josephine Rodrigues, M.D., Ph.D.

Academic Editor

Journal Requirements:

1. We have noticed that you have uploaded Supporting Information files, but you have not included a list of legends. Please add a full list of legends for your Supporting Information files after the references list. 

Additional Editor Comments (if provided):

Reviewers' comments:

Reviewer's Responses to Questions

**Comments to the Author**

1. Does this manuscript meet PLOS Global Public Health’s publication criteria? Is the manuscript technically sound, and do the data support the conclusions? The manuscript must describe methodologically and ethically rigorous research with conclusions that are appropriately drawn based on the data presented.

Reviewer #1: Yes

Reviewer #2: Yes

Reviewer #3: Partly

2. Has the statistical analysis been performed appropriately and rigorously?

Reviewer #1: No

Reviewer #2: Yes

Reviewer #3: I don't know

3. Have the authors made all data underlying the findings in their manuscript fully available (please refer to the Data Availability Statement at the start of the manuscript PDF file)?

Reviewer #1: Yes

Reviewer #2: Yes

Reviewer #3: Yes

4. Is the manuscript presented in an intelligible fashion and written in standard English?

Reviewer #1: Yes

Reviewer #2: Yes

Reviewer #3: Yes

5. Review Comments to the Author

Reviewer #1: This is a well-written manuscript on a subject of very high interest in the field of tuberculosis. Local authorities and policymakers will benefit from this publication, and the WHO End TB strategies will benefit from this knowledge.

Some remarks:

The authors should provide more data from their country or region on the proportion of TB cases in prisons compared to the total number of cases in the country. Furthermore, it might be interesting to compare the number of TB cases that existed before in those five prisons, and how much was collected in a year through active case-finding. One way to measure the impact of this intervention and estimate the costs of implementing it periodically is to use the following method.

The title "New Admissions and Asymptomatic TB Cases fuel TB Epidemic in Prisons, a 2 Cross Sectional Survey in Tanzania" is questionable. I would suggest being less definitive, and using the phrase "seems to fuel" instead of "fuel".

Line 62, Please review current literature such: Martinez L et al “Global, regional, and national estimates of tuberculosis incidence and case detection among incarcerated individuals from 2000 to 2019: a systematic analysis.”

Line 90. Screening was performed upon entry or transfer from the prison. How this was done can be more detailed. This is because it is not clear what is done on the day of entry or up to six months after entering prison. It would be good to know how much time, on average, it took to perform the analysis on a new admission.

Line 96. The questionnaire collected information on the date of admission to prison and included the date of confirmation of TB. Therefore, we have the necessary tools to perform Cox regression. Why did the authors decide to perform logistic regression?

Line 167. These lines suggest that Ukonga prison does not receive new admissions, and Segerea does not receive convicted criminals. This also suggests that among the five prisons, the dynamics of admission and mobility are very different. Therefore, specific analyses of the variables for each prison are required. Can the authors comment on this?

Table 2 contains the same data as in Table 1, with only a difference in how the percentage is displayed. My suggestion is to eliminate Table 1 and change it to a flowchart that explains the steps of the study about the people who took questionnaires, how many refused to participate, how many took the Xpert, then which ones were valid, and finally how many cases had TB.

Table 2. In the methodology, the authors state that the final model only includes variables with a p-value <0.1. But if there is the variable “number of cellmates” (p value:0.195). If all variables are going to be included, consider also important to include the “New admission” variable in Table 2. This is mainly because, owing to the chosen title, it is an important variable.

Line 186. TB symptoms were the best predictors of TB (4.07 vs 1.06). However, 57% (122/214) of the diagnosed patients had no symptoms. This is perhaps the most important aspect of the present study. Therefore, the author must discuss it more and perhaps review some limitations in the collection of symptoms, knowledge, language barriers, and so on.

Line 188. The authors say this: “Therefore, an effective screening intervention is one that applies mixed methods that detect both symptomatic and asymptomatic tuberculosis.” Have you discussed how often it should be done, the approximate cost, and what would you recommend for LIC and LMIC countries? There is literature on this, mainly from Brazil. This is discussed in the conclusion section.

Line 190- To review. There were 9077 new admissions, and 132 (1.45%) had TB. Prisoners under 6 months of age comprised 62.3% (8643/13763) of the total population. This group comprised 53.3% (114/214) of the TB cases. New admissions have a lower prevalence of TB than previous prisoners. In absolute numbers, there are more because 66% of the population came from a new admission. However, in terms of rates, it was lower. The risk of developing tuberculosis increases over time in prisons. The TB notification rate of the group of those who spent less time was lower when compared to those who spent more time in prison and below the global rate found (1555/100 thousand inhabitants). Therefore, I conclude that the incidence rate is very similar in all groups, but increases with time in prison and with the number of readmissions.

It is a significant aspect to note that the prisoner turnover is quite high. Approximately 2 times its static size. The median time spent in prison was 1.5 months. This means that a large part of the population is always new, and everything that occurs in it will be the majority. Nevertheless, in terms of relative comparison, it does not hold true.

It is important and appropriate for authors to address certain limitations of the study.

Reviewer #2: Dear Author,

Thank you for sending the manuscript. These are some of the observations and suggestions made based on my knowledge and understanding of the paper. Please check and do the needful.

Abstract

28: No mention of early morning samples in the main text

36: Error in the calculated prevalence, please correct

Introduction

52-54 : Please add reference

55: Updated statistics available, please add the latest numbers

72-75: Split the long sentence into two for better understanding and mention if it is universal screening for all new inmates irrespective of whether they are symptomatic or not?

* All references needs to be in square brackets according to the author guidelines of journal

Methods

90-92: Please define new admission to prison and on an average how many days after enrolment were they screened if that data is available

94: Is the screening tool used a validated questionnaire?

95: Please remove full stop in between the sentence

99: Please explain why two sputum samples were not collected for all according to the diagnostic guidelines

*There is no mention of the minimum required sample size for the study. Please include sample size calculation also and the reference for the same

Ethical consideration

*Better to be mentioned before the study procedures

Results

141: It would be more explanatory if the total number of prisoners approached, number consented and number screened is added to know the missed numbers as well.

142: out of 16312 consented, why only 13868 were tested? What about the remaining 2264?

154:

Table1

Number of cellmates: total numbers have an error

Prison ID:Please check the prevalence rates, disparity noted.

174:

Table 2

Number of incarcerations :spelling mistake

Discussion:

177: "more than" "higher than" both phrases mentioned in the same sentence

180-183:Split the sentence for better understanding

Thank you

Reviewer #3: Authors are presenting important findings from a multi-site active TB case finding survey in Tanzanian correctional system. Below are recommendations to make the manuscript suitable for publication:

OVERALL

=====

- The manuscript would benefit from a thorough proof-reading

- Authors are advised to use a humanising language for people in incarceration. We are trying to avoid "inmates" or "prisoners" and replace them with "incarcerated individuals" or "people who are incarcerated". Please have a look though this article. Tran, N.T., Baggio, S., Dawson, A. et al. Words matter: a call for humanizing and respectful language to describe people who experience incarceration. BMC Int Health Hum Rights 18, 41 (2018). https://doi.org/10.1186/s12914-018-0180-4

ABSTRACT

=======

- Lacks a sentence describing the aims of the study

- Provide TB case definition

INTRODUCTION

- Proper review of relevant literature is required

- Provide local (Tanzanian) TB data, including available TB in prison information

METHODS

======

- Describe how the already incarcerated individuals were recruited? It seems that only half of this population was recruited (4000 out of 8000 individuals). This attracts a potentially major source of selection bias, if particular individuals were selected or only those presented to prisons' clinics!

- Similarly, describe why these prisons in particular were chosen? randomly? What is so special about them?

- The use of prison staff in administering the questionnaire and collecting the sputum specimens might have jeopardised the confidentiality of participants and caused a breach of your ethical statement. Can you elaborate further on this!

- It is very important to describe the TB symptoms collected! What were they?

- Describe whether the sputum handling was done in accordance with the manufacturer's guidelines.

- Justify the categorisation of each variable to the described categories. e.g age, times of incarceration...etc.

- It would be very good to the model to adjust findings to prisons, instead of using prisons as a variable.

- More information is required on the multivariable regression analysis. Did you use forward/backward handling of variables? How did you choose the final model?

RESULTS

======

- It is not possible to consider the finding for the study as the prevalence of TB (or sometimes you use the burden of TB) in these prisons, unless you further explain the recruitment process of the already incarcerated individuals. If is not involving ALL, then you can simply say "prevalence of TB in the screened sample", which can be the closest to a representative sample.

- Can you further describe the meaning to the "duration of incarceration"? Is the sentence duration or the period spent already in prison.

DISCUSSION

========

- No section describing the limitations of the study!

- Limited description of the findings.

- Needs further review of the literature to add relevant details.

- The use of symptoms to define TB cases in prisons might have poor utility and might underestimate the true prevalence of Tb (See PMID: 27197601). In your study more than half (N=112) of TB cases would be missed if relied on TB symptoms.

CONCLUSION

=========

- As above, further information is required in order to consider the finding as the prevalence of TB in the prison system.

6. PLOS authors have the option to publish the peer review history of their article (what does this mean?). If published, this will include your full peer review and any attached files.

**Do you want your identity to be public for this peer review?** For information about this choice, including consent withdrawal, please see our Privacy Policy.

Reviewer #1: No

Reviewer #2: No

Reviewer #3: **Yes: **Haider Al-Darraji

---

## [Decision Letter · Decision Letter 1]

16 Feb 2024

PGPH-D-23-01618R1

New Admissions and Asymptomatic TB Cases Seem to fuel TB Epidemic in Prisons, a Cross Sectional Survey in Tanzania

Dear Dr. Mangu,

Thank you for submitting your manuscript to PLOS Global Public Health. After careful consideration, we feel that it has merit but does not fully meet PLOS Global Public Health’s publication criteria as it currently stands. Therefore, we invite you to submit a revised version of the manuscript that addresses the points raised during the review process.

EDITOR: Kindly address comments from reviewers, with appropriate rebuttal/justification if the comments were not incorporated.

We look forward to receiving your revised manuscript.

Kind regards,

Rashmi Josephine Rodrigues, M.D., Ph.D.

Academic Editor

Journal Requirements:

Additional Editor Comments (if provided):

Reviewers' comments:

Reviewer's Responses to Questions

**Comments to the Author**

1. If the authors have adequately addressed your comments raised in a previous round of review and you feel that this manuscript is now acceptable for publication, you may indicate that here to bypass the “Comments to the Author” section, enter your conflict of interest statement in the “Confidential to Editor” section, and submit your "Accept" recommendation.

Reviewer #2: All comments have been addressed

Reviewer #3: (No Response)

2. Does this manuscript meet PLOS Global Public Health’s publication criteria? Is the manuscript technically sound, and do the data support the conclusions? The manuscript must describe methodologically and ethically rigorous research with conclusions that are appropriately drawn based on the data presented.

Reviewer #2: Yes

Reviewer #3: Partly

3. Has the statistical analysis been performed appropriately and rigorously?

Reviewer #2: Yes

Reviewer #3: Yes

4. Have the authors made all data underlying the findings in their manuscript fully available (please refer to the Data Availability Statement at the start of the manuscript PDF file)?

Reviewer #2: Yes

Reviewer #3: Yes

5. Is the manuscript presented in an intelligible fashion and written in standard English?

Reviewer #2: Yes

Reviewer #3: Yes

6. Review Comments to the Author

Reviewer #2: Thank you for replying to the comments and making the suggested edits.

These are some observations on the comments:

94- Standard screening tools are generally used for screening for a particular disease/ condition.It has sensitivity and specificity. If you have made a questionnaire with questions prepared by yourselves, it would be great if you pilot test and face validate the questionnaire with the help of an expert.If this is followed, please mention it in the manuscript or mention it as a limitation of the study

99- It cannot be mentioned as an intervention study: screening for a disease and doing lab investigations to confirm the screen positives is not intervention.

Sample size calculation: The ultimate goal of achieving optimum sensitivity in terms of power and confidence in an experiment requires consideration of the optimum number of subjects included(i.e. minimum sample size.

Sampling technique :the sampling technique that you are using in the selection of your estimated sample size is different in different scenarios based on the appropriateness. Hence this may not be a correct explanation for the same.

Hence its always better to calculate the minimum sample size.

141- In the flow chart you have mentioned, 2264 excluded because they were not able to produce sputum. explain "105 invalid or error" in what?-sputum sample collection? Please incorporate the necessary details written in the text lines 153-156 to methodology flow chart so that it is self-explanatory.

Please include the limitations of the study considering all the factors reviewers have commented.

All the best!

Reviewer #3: I have noticed that the manuscript has improved, but most of my comments on the earlier version were answered in the reply letter, but recommendations were not incorporated in the text. Comments made by the reviewers are meant to improve the manuscript and make it suitable for publications not to be answered in a reply letter.

For example:

- Please incorporate in the text that only half of the approached incarcerated individuals consented for the screening.

- Please include in the limitations that the use of prison's healthcare worker might attract a confidentiality breach, but for logistic (?) reasons, they were used! Even if they were trained on ethical principles, they remain a source of confidentiality breach, given their representation of the prison department.

- Explicitly present the TB symptoms you described in the letter inside the manuscript text.

- My question about the categorisation is not about the choice of variables, but rather the numbers used in each category cells. ? have equal numbers, ? using median... etc

- Change variable of duration of incarceration to "time spent in prison", the first one is misleading as it might imply what they were sentenced with, which is different.

7. PLOS authors have the option to publish the peer review history of their article (what does this mean?). If published, this will include your full peer review and any attached files.

**Do you want your identity to be public for this peer review?** For information about this choice, including consent withdrawal, please see our Privacy Policy.

Reviewer #2: No

Reviewer #3: **Yes: **Haider Al-Darraji

---

## [Decision Letter · Decision Letter 2]

10 Jul 2024

PGPH-D-23-01618R2

New Admissions and Asymptomatic TB Cases Seem to fuel TB Epidemic in Prisons, a Cross Sectional Survey in Tanzania

Dear Dr. Mangu,

Thank you for submitting your manuscript to PLOS Global Public Health. After careful consideration, we feel that it has merit but does not fully meet PLOS Global Public Health’s publication criteria as it currently stands. Therefore, we invite you to submit a revised version of the manuscript that addresses the points raised during the review process.

EDITOR: Please make minor revision as suggested by the reviewer.

We look forward to receiving your revised manuscript.

Kind regards,

Rashmi Josephine Rodrigues, M.D., Ph.D.

Academic Editor

Journal Requirements:

Additional Editor Comments (if provided):

Reviewers' comments:

Reviewer's Responses to Questions

**Comments to the Author**

1. If the authors have adequately addressed your comments raised in a previous round of review and you feel that this manuscript is now acceptable for publication, you may indicate that here to bypass the “Comments to the Author” section, enter your conflict of interest statement in the “Confidential to Editor” section, and submit your "Accept" recommendation.

Reviewer #2: All comments have been addressed

Reviewer #3: All comments have been addressed

2. Does this manuscript meet PLOS Global Public Health’s publication criteria? Is the manuscript technically sound, and do the data support the conclusions? The manuscript must describe methodologically and ethically rigorous research with conclusions that are appropriately drawn based on the data presented.

Reviewer #2: Yes

Reviewer #3: Yes

3. Has the statistical analysis been performed appropriately and rigorously?

Reviewer #2: Yes

Reviewer #3: Yes

4. Have the authors made all data underlying the findings in their manuscript fully available (please refer to the Data Availability Statement at the start of the manuscript PDF file)?

Reviewer #2: No

Reviewer #3: Yes

5. Is the manuscript presented in an intelligible fashion and written in standard English?

Reviewer #2: Yes

Reviewer #3: Yes

6. Review Comments to the Author

Reviewer #2: 41- Since the prevalence of TB was more among the inmates compared to new admissions

(as mentioned in the results) and there is no significant association found between duration of length of stay (if new admissions is defined as <6months) the statement "new admissions could be the driving force of TB epidemic" in the conclusion of abstract cannot be made. Please add the answer to your objective as the conclusion.

24,62,178- Prevalence of TB in Tanzania/prisons is mentioned as prevalence per lakh population as well as in person years in the abstract , introduction and discussion. I think its better to use the same unit throughout.

Reviewer #3: Authors have addressed comments raised previously.

7. PLOS authors have the option to publish the peer review history of their article (what does this mean?). If published, this will include your full peer review and any attached files.

**Do you want your identity to be public for this peer review?** For information about this choice, including consent withdrawal, please see our Privacy Policy.

Reviewer #2: No

Reviewer #3: **Yes: **Haider Al-Darraji

---

## [Decision Letter · Decision Letter 3]

12 Sep 2024

New Admissions and Asymptomatic TB Cases Seem to fuel TB Epidemic in Prisons, a Cross Sectional Survey in Tanzania

PGPH-D-23-01618R3

Dear Dr Mangu,

We are pleased to inform you that your manuscript 'New Admissions and Asymptomatic TB Cases Seem to fuel TB Epidemic in Prisons, a Cross Sectional Survey in Tanzania' has been provisionally accepted for publication in PLOS Global Public Health.

Best regards,

Rashmi Josephine Rodrigues, M.D., Ph.D.

Academic Editor

Reviewer Comments (if any, and for reference):

Reviewer's Responses to Questions

**Comments to the Author**

1. If the authors have adequately addressed your comments raised in a previous round of review and you feel that this manuscript is now acceptable for publication, you may indicate that here to bypass the “Comments to the Author” section, enter your conflict of interest statement in the “Confidential to Editor” section, and submit your "Accept" recommendation.

Reviewer #2: All comments have been addressed

2. Does this manuscript meet PLOS Global Public Health’s publication criteria? Is the manuscript technically sound, and do the data support the conclusions? The manuscript must describe methodologically and ethically rigorous research with conclusions that are appropriately drawn based on the data presented.

Reviewer #2: Yes

3. Has the statistical analysis been performed appropriately and rigorously?

Reviewer #2: Yes

4. Have the authors made all data underlying the findings in their manuscript fully available (please refer to the Data Availability Statement at the start of the manuscript PDF file)?

Reviewer #2: Yes

5. Is the manuscript presented in an intelligible fashion and written in standard English?

Reviewer #2: Yes

6. Review Comments to the Author

Reviewer #2: Dear Author,

Thank you for incorporating all the changes suggested. The manuscript is good for publication.

Small comment: Introduction- all paragraphs are starting with the word "despite"- it would be great if you could change any of it.

All the best!

7. PLOS authors have the option to publish the peer review history of their article (what does this mean?). If published, this will include your full peer review and any attached files.

**Do you want your identity to be public for this peer review?** For information about this choice, including consent withdrawal, please see our Privacy Policy.

Reviewer #2: No
